# Role of Tumor-Associated Neutrophils in the Molecular Carcinogenesis of the Lung

**DOI:** 10.3390/cancers13235972

**Published:** 2021-11-27

**Authors:** Elisabeth Taucher, Valentin Taucher, Nicole Fink-Neuboeck, Joerg Lindenmann, Freyja-Maria Smolle-Juettner

**Affiliations:** 1Department of Internal Medicine, Division of Pulmonology, Medical University of Graz, 8010 Graz, Austria; 2Department of Internal Medicine, Division of Cardiology, Hospital Barmherzige Schwestern Ried, 4910 Ried, Austria; valentin.taucher@medunigraz.at; 3Department of Thoracic Surgery, Medical University of Graz, 8010 Graz, Austria; nicole.neuboeck@medunigraz.at (N.F.-N.); jo.lindenmann@medunigraz.at (J.L.); freyja.smolle@medunigraz.at (F.-M.S.-J.)

**Keywords:** tumor-associated neutrophils, inflammation, microenvironment, lung cancer

## Abstract

**Simple Summary:**

This review of the literature aims at giving a concise overview of the impact of tumor-associated neutrophils (TANs) on lung carcinogenesis. In the first part of this manuscript, the general action mode of TANs in cancer is depicted, listing studies on several cancer entities and on mouse models. The latter part of this work focuses specifically on TANs in lung cancer, giving an outlook on future therapeutic implications of cancer immunity, using, for example, anti-cancer vaccines.

**Abstract:**

Tumorigenesis is largely influenced by accompanying inflammation. Myeloid cells account for a significant proportion of pro-inflammatory cells within the tumor microenvironment. All steps of tumor formation and progression, such as the suppression of adaptive immune response, angio- and lymphangiogenesis, and the remodeling of the tumor stroma, are to some degree influenced by tumor-associated immune cells. Tumor-associated neutrophils (TANs), together with tumor-associated macrophages and myeloid-derived suppressor cells, count among tumor-associated myeloid cells. Still, the exact molecular mechanisms underlying the tumorigenic effects of TANs have not been investigated in detail. With this review of the literature, we aim to give an overview of the current data on TANs, with a special focus on lung cancer.

## 1. Introduction

The process of tumorigenesis is strongly influenced by accompanying inflammation [1]. Myeloid cells constitute a significant proportion of all pro-inflammatory cells, which the tumor microenvironment harbors [2]. The suppression of adaptive immune responses and the endorsement of angiogenesis and lymphangiogenesis, as well as the remodeling of the tumor stroma, are influenced by tumor-infiltrating immune cells, thereby influencing—to some degree—every step of tumor progression and metastasis [3]. In the past, more research has been done on tumor-associated macrophages than on tumor-associated neutrophils (TANs). However, in the last decade, TANs have been increasingly investigated and their underlying molecular mechanisms have been gradually elucidated.

## 2. General Aspects

There have been studies on mice about TANs, revealing their pivotal role, namely having protumor as well as anti-carcinogenic properties [4,5,6]. Hence, the concept of anti-carcinogenic “N1 neutrophils” and pro-carcinogenic “N2 neutrophils” has been established based on the findings from murine models [7]. Still, it remains unclear whether this paradigm can be translated to human tumor biology.

Lung cancer, in spite of the development of novel immunotherapeutic agents and other evolving treatment strategies, remains the number one cause of cancer-associated deaths globally [8,9]. Tobacco smoking is the most common risk factor for lung cancer; however, 10–15% of cases are not related to smoking or any other common risk factor [10]. Aside from the importance of preventive measures, an accurate understanding of molecular lung carcinogenesis—and immunological aspects, in particular—is crucial for developing new, effective treatment agents. The lung cancer tumor microenvironment has become a focus of research lately. Understanding the complex impact of tumor-infiltrating immune cells will therefore be of major importance in order to gain more insight into lung carcinogenesis.

With this review of the literature, we aim to give an overview of the current data on TANs, with a special focus on their role in lung cancer.

### 2.1. Tumor-Associated Neutrophils—What Do We Know?

In general, the tumor microenvironment is made up of multiple types of immune cells, comprising macrophages, dendritic cells, natural killer cells, T cells, and B cells, as well as neutrophils. It has become clear that the interplay of these cells impacts tumor development significantly [11,12,13]. TANs make up a vast proportion of immune cells in many cancer entities, as has been shown in breast, gastric, renal, and lung cancers, as well as in malignant melanoma [7,14]. According to the existing literature, TANs function in complex and versatile ways, exhibiting direct cytotoxicity towards cancer cells, inhibiting metastasis, but also being capable of exerting pro-carcinogenic effects [7,15,16,17]. Pro-carcinogenic effects comprise neo-angiogenesis, the promotion of cancer cell motility and epithelial-to-mesenchymal transition, and the migration, invasion, and modulation of other immune cells, called the “immunosuppressive switch” [18,19]. The peculiarity of TANs actually lies in their aberrant activation, as recent studies have demonstrated: TANs undergo alternative activation, dependent on a variety of cues they are exposed to within the tumor microenvironment [20,21].

Neutrophils as primary inflammatory cells are crucial protectors of the host during the early phase of microbial infections [4]. In cancers, a high functional plasticity of neutrophils has been shown, suggesting a rather pro-tumor effect of those neutrophils which are related to granulocytic myeloid-derived suppressor cells. Brandau et al. proposed a model of TANs wherein they are constantly recruited and activated in tumors, acting as mainly pro-carcinogenic [4]. On the other hand, tumor therapies have also been shown to acutely activate neutrophils, which then act synergistically with anti-tumor therapeutics. Thus, a future therapeutic option for cancers might be the functional conversion of pro-carcinogenic TANs to anti-tumorigenic ones. In human immunohistochemical studies, TAN infiltration was associated with a poor outcome in head and neck cancer [22], renal cancer [23], melanoma [24], liver cancer [25], and colon cancer [26]. Conversely, in patients suffering from gastric cancer, a high proportion of TANs was linked to a better outcome [27]. Caruso et al. conducted this study on gastric cancer because until then the evidence suggested neutrophils act nonspecifically in cancers. A total of 273 patients suffering from late-stage gastric carcinoma were enrolled in the study and the number of TANs was investigated and correlated with survival rates. Female patients in this study, who were found to have a moderate-to-high amount of TANs, showed a 39% reduction in their mortality risk. Interestingly, no such association was found in male patients. The authors suggested gender differences in host-defense mechanisms and specifically in TAN function as being responsible for this finding [27].

The controversial data on the function of TANs in tumors can partly be explained by their distinct localization within a given solid tumor. Indeed, not only the presence or absence of TANs in tumors but also their localization is of prognostic importance [28,29,30,31,32]. Dependent on whether TANs are located within the tumor, adjacent to it, or in the stroma, the prognostic implications are different. Most studies have shown that intratumoral, as compared to peritumoral or stromal TANs, are most closely linked to an adverse outcome [19]. However, in a few studies, a link between peritumoral TANs and a poor outcome has also been observed. In hepatocellular cancer, for instance, many studies have shown that TANs, if they do occur, are usually located in the stroma adjacent to the tumor and not directly within the tumor. These stromal TANs are indicators of an adverse prognosis in hepatocellular cancer [31,32,33]. In a trial on patients suffering from cervical cancer, an elevated count of CD66b+ TANs within the tumor stroma was an indicator of a shorter recurrence-free survival. Yet, TANs directly located in tumor nests were not [29]. The same study by Carus et al. also demonstrated the whole-tumor TAN-to-CD8+ ratio as being a good marker for the prediction of recurrence-free survival. 

A large abundance of neutrophils located directly in cancer nests has been strongly associated with metastasis, late-stage disease, and a shorter overall and disease-free survival, as shown by a study on esophageal cancer patients [34]. Still, the same study showed that a high ratio of peritumoral neutrophils to CD8+ lymphocytes was associated with more advanced-stage cancers and the presence of lymph node invasion [34].

These partly controversial results on the effect of TANs on patients’ outcomes leave much room for future research. The current literature indicates that the mode of action of neutrophils is largely dependent on their activation status and location, but also differs considerably between tumor specimens.

### 2.2. Murine Models of TANs and Potential Translation to Human Cancers

Neutrophils constitute a major proportion of circulating white blood cells, acting as a first-line defense against microbial infection or tissue injury [35,36]. Neutrophils are recruited from the vasculature to tissues by chemokines, which then activate neutrophils for conquering microbes or initiating tissue repair mechanisms [37]. A dysregulation in chemokine–neutrophil crosstalk has been linked to a variety of diseases, including several cancer entities [35,38,39]. Neutrophils, in the context of an inflammatory microenvironment, recruit macrophages, which in turn influence neutrophil function. The presence of TANs, as well as their recruitment and activation, play an important role in the maintenance of the tumor microenvironment and tumor progression [40].

First and foremost, it must be pointed out that immunologic features differ considerably between humans and mice, as has been demonstrated by several studies with negative results [41,42,43,44]. There is no solid systematic evidence supporting the assumption that human and murine immunology are largely interchangeable [45]. Genome-wide expression analyses of white blood cells from humans after severe trauma or burn injuries and analogies of samples from well-established mouse models of trauma and burns allow for a comparison between the pro-inflammatory genes in humans and mice [46,47,48,49,50]. A review article on these genome-wide analyses by Seok et al., however, showed a rather poor correlation of immunological responses between humans and mouse models [45].

Only a few studies on mouse models and the immunological aspects of cancer have been carried out. When looking at clinical studies with patients suffering from different cancers, the literature suggests a mainly pro-carcinogenic effect of TANs in humans [5]. TANs are observed in or nearby nearly all solid tumors and may secrete reactive oxygen species and proteinases capable of modifying tumor growth and invasiveness. Human studies have shown that many cancer entities recruit neutrophils to the site of tumorigenesis, enhancing tumor progression [5].

As mentioned above, there is a pivotal role of TANs in carcinogenesis, as has been indicated by means of murine models and in humans [7]. In mice, TANs evidently have pro-carcinogenic as well as anti-carcinogenic effects [4,5,6]. Given the fact that both the innate and adaptive immune systems differ considerably between humans and other species, the molecular pathways leading to this pivotal role of TANs and the exact underlying mechanisms are not likely to be interchangeable in mice and men, however.

Pylaeva et al. investigated the properties of TANs using transplantable tumor models [51]. The authors were able to show that nicotinamide phosphoribosyltransferase (NAMPT) impacts the downstream signaling of CSF3R, a cytokine responsible for the generation, differentiation, and function of neutrophils. NAMPT influenced the conversion of TANs to a pro-carcinogenic and pro-angiogenic phenotype. Upon the conversion of the TANs, tumor vascularization and growth were enhanced. The inhibition of NAMPT led to the conversion of TANs to an anti-carcinogenic phenotype. The anti-carcinogenic TANs were transferred into B16F10-tumor-bearing mice, and an attenuation of tumor angiogenesis and progression was observed. Notably, NAMPT is up-regulated in TANs from human skin cancer as well as head and neck cancer, and NAMPT expression positively correlates with the tumor stage. Hence, the authors of this study concluded that the regulation of NAMPT signaling and its effect on TANs is transferrable from the murine model to human cancers [51].

One interesting study on mice showed, that neutrophil-derived IL-1β blocks the effect of nuclear factor κB (NF-κB) inhibitors against lung cancer [52]. It is well known that NF-κB signaling is part of tumor development and progression in lung cancer. However, NF-κB inhibiting agents lack effectiveness in lung cancer treatments. The authors of this study aimed to explain this paradox by means of a mouse model, with the genetic deletion of the inhibitor of NF-κB kinase subunit beta (IKKβ) in myeloid cells [52]. In KRAS-driven lung cancer, the IKKβ-depleted mice showed enhanced tumorigenesis. When IL-1β processing by cathepsin G was increased in TANs by the myeloid-specific inhibition of NF-κB, this led to the enhancement of epithelial cell proliferation. Following myeloid-specific NF-κB inhibition, the NF-κB inhibitor bortezomib reduced tumor formation and tumor growth in the mice [52]. These findings highlight the importance of TANs in the context of IL-1β in lung carcinogenesis. Since, in human lung cancer patients, plasma IL-1β levels correlate with a poor prognosis, and bortezomib treatment led to an increase in IL-1β levels [52], the above-mentioned mouse study can be translated to human lung cancer as well.

Vikis et al. also carried out a mouse study on the immunological aspects of lung carcinogenesis [53]. The hypothesis of this study was that neutrophils play a role in methylcholanthrene (MCA)-initiated butylated hydroxytoluene (BHT)-promoted lung carcinogenesis in mice. Interestingly, it has been observed in the past that the susceptibility of mice strains to MCA-initiated BHT-promoted lung cancer is variable [53]. In tumor-susceptible BALB/cByJ (BALB) mice, the neutrophil count in the lungs was significantly elevated when compared to tumor-resistant C57BL/6J (B6) mice. Treatment with BHT led to a further increase in neutrophil number, and neutrophil depletion in BALB mice led to a 71% decrease in tumor growth rate. Of note, the T cell count was not associated with tumor multiplicity, although T cells also increased upon BHT treatment. This mouse study also highlights the important role of TANs in lung cancer, which is clearly different from the role of tumor-associated T cells [53].

Another study aimed at investigating the immune compartment of tumors in greater detail, with a special focus on TANs [54]. Here, a Kras (G12D)-driven murine model of lung cancer was used to outline an immune signature and the exact role GR1+ neutrophils play in cancer progression. In GR1+ depleted mice, tumor growth was attenuated, T cell homing was enhanced, and anti-PD1 immunotherapy was more effective. Moreover, angiogenesis was significantly influenced by GR1+ neutrophils, resulting in tumor hypoxia and enhanced expression of Snail. In turn, Snail contributed to a quicker progression of the disease, endorsing the further infiltration of neutrophils [54]. These results clearly indicate a pro-tumorigenic function of GR1+ neutrophils, and the authors even proposed the interplay of Snail and TANs to be a vicious cycle, rendering the tumor microenvironment increasingly carcinogenic.

Furthermore, a study was conducted using single-cell RNA sequencing to map neutrophils among other tumor-infiltrating myeloid cells in patients suffering from NSCLC [55]. The authors pointed out that the complex diversification and activation process of tumor-infiltrating myeloid cells is generally poorly understood. TANs undergo various states of activation, and each state is associated with a different mode of function. In the NSCLC patients, 25 different states of tumor-associated myeloid cells were outlined and were found to be reproducible between patients. Tumor-associated myeloid cells were profiled in mice as well, and, notably, a near-perfect congruence of conserved neutrophil subsets was found between mice and humans. The same congruence held true for dendritic cells, monocytes, and macrophages. When comparing myeloid cells from the patients’ blood samples, however, only a minor overlap was found in the cell population structures of tumor-infiltrating myeloid cells. This study by Zilionis et al. shows the reproducible congruence between TANs in mice as well as in humans, which is not found in neutrophils from blood samples [55].

Granot et al. conducted a mouse study focusing specifically on the implications of TANs in the premetastatic lung [17]. Since primary cancers have the capability to prepare potential metastatic sites for colonization and consecutive metastatic growth, the hypothesis was that TANs also impact this process. Indeed, in their mouse model of 4T1 breast cancer cells orthotopically implanted into mice, the authors of this study could demonstrate that TANs inhibit metastatic spread to the lungs by the generation of H2O2. Moreover, CCL2 secreted by the tumor renders the lung microenvironment anti-metastatic by recruiting G-CSF-stimulated neutrophils. To summarize, a pivotal role of the tumor-secreted factors of cancer immunity was discovered: on the one hand, these soluble factors endorse tumor progression at the primary site, but on the other hand, they inhibit the metastatic process in distant organs [17].

In mice, some evidence suggests also an anti-carcinogenic effect of TANs. However, research now focuses rather on the inhibition of pro-carcinogenic substances secreted by TANs, since, in humans, the tumor-promoting effect of TANs seems to outweigh any tumor-suppressive properties.

Figure 1 shows the possible mechanisms by which established anti-tumor treatments might modulate the phenotype and action mode of TANs. Figure adapted from Shaul et al. [56].

## 3. TANs in Lung Cancer

The role of TANs in lung cancer has not yet been elucidated in great detail. One study from 2013 was performed to investigate the prognostic significance in non-small cell lung cancer (NSCLC) [29]. A total of 335 patients who had undergone surgery for stage I–IIIA NSCLC were enrolled into this analysis and assessed for CD66b(+) neutrophils and CD163(+) macrophages in tumor nests and the adjacent stroma by means of immunohistochemistry. When correlating the presence of neutrophils and macrophages with the patients’ outcomes, a higher density of CD66b(+) neutrophils, intratumoral as well as in the tumor stroma, was associated with elevated levels of C-reactive protein and a higher white blood cell count [22]. However, neither the abundance of CD66b(+) neutrophils nor of CD163(+) macrophages were significantly linked to recurrence-free survival or overall survival (OS) [29].

In one study, the phenotype and function of TANs, specifically in early-stage lung cancer, were investigated on surgically obtained tumor tissue [2]. One goal of this study was to elucidate the interaction of activated neutrophils with T cells within the tumor stroma. TANs can regulate the T cell response, and cytotoxic T-lymphocytes whose function is impacted by TANs may exert antigen-driven anti-tumor immunity. According to previous studies, TANs can present antigens, thereby triggering T cell activation [57,58,59,60]. The tumor-suppressive effects of neutrophils in cancer patients are most probably linked to circulating low-density granulocytic myeloid-derived suppressor cells (G-MDSCs) [61,62]. There is still uncertainty, however, whether the presence of G-MDSCs is linked to a complex host-versus-tumor response of the immune system or is simply a consequence of disease progression. The results of the above-mentioned study on the function of TANs in early-stage lung cancer [2] showed that TANs constituted 5–25% of the cells isolated from the cancer samples. When the TANs were compared to blood-derived neutrophils from patients, they showed an activated phenotype and harbored chemokine receptors including CCR5, CCR7, CXCR3, and CXCR4. Moreover, TANs in these early-stage lung cancer samples secreted significant amounts of pro-inflammatory cytokines such as IL-6 and IL-8. Interestingly, not only TANs but also neutrophils from healthy lung tissue were able to stimulate T cell proliferation and interferon-γ release [2]. Hence, a complex cross-talk between TANs and activated T cells was found to upregulate certain molecules on the neutrophil surface, including CD54 and CD86, endorsing T-cell proliferation. The conclusion drawn from this study is that TANs in early-stage lung cancer are immune-stimulating and trigger an anti-tumor T-cell response.

In a study by Ilie et al. from 2012, the prognostic relevance of the intratumoral cluster of differentiation 66b (carcinoembryonic antigen-related cell adhesion molecule 8 (CD66b))-positive neutrophils and of the intratumoral CD66b-positive neutrophil-to-cluster of differentiation 8 (cell surface antigen T8 (CD8))-positive lymphocytes (the CD66b-positive neutrophil-to-CD8-positive lymphocyte ratio (iNTR)) was evaluated in patients with NSCLC [30]. In total, 632 samples of resectable lung cancer were included in this analysis. It was found that the proportion of CD66b-positive TANs within tumors was elevated in 50% of patients. A high amount of CD66b-positive neutrophils went along with a worse prognosis, indicating a higher probability of relapse and poorer overall survival (median OS, 57 months for low CD66b-positive cell density; 54 months for high CD66b-positive cell density; *p* = 0.088) [30]. The iNTR was higher than normal in only 30% of patients and was strongly linked to a high incidence of relapse and a poor prognosis, as well (median OS: 60 months for an iNTR ≤ 1; 46 months for an iNTR > 1; *p* < 0.0001) [23]. A high relapse probability was linked to a high iNTR in the multivariate analysis, as well, and the iNTR was outlined as an independent prognostic factor indicating a worse overall survival [30].

When closely looking at myeloid cells in NSCLC, one study showed that certain neutrophil transcription signatures were predictors of mortality [63]. In this analysis, the molecular profiles of cancers and cells from the cancer microenvironment were evaluated. A meta-analysis of expression signatures from ∼18,000 human cancer samples, comprising 39 malignancies, including NSCLC, was conducted with respect to OS. A computational approach to the tumor transcriptome, using the software CIBERSORT, revealed 22 specific leukocyte subsets associated with survival [63]. TAN signatures, but also plasma cell signatures, specifically predicted survival in lung adenocarcinomas, highlighting the important role of TANs in tumorigenesis. Another study showed that endoplasmic reticulum stress-associated inositol-requiring protein 1 (IRE1; also known as ERN1)-X-box binding protein 1 (XBP1) signaling converts neutrophil function toward a highly immunosuppressive phenotype [64,65]. Generally, in NSCLC, the LOX1+ phenotype of myeloid-derived suppressor cells (MDSCs) is increased. Therefore, future studies may reveal the role of LOX1+ MDSCs as prognostic biomarkers in NSCLC [65]. The elimination of LOX1+ MDSCs by means of targeted therapeutics may be a novel tool in future NSCLC treatment.

In human NSCLC, neutrophils were shown to make up the majority of immune cells within and adjacent to the tumor [66]. A subset of TANs has recently been identified, expressing markers of both neutrophils and antigen-presenting cells, so these specific TANs can cross-present antigens and enhance anti-tumor T cell responses [67]. Moreover, IL-1β is expressed by neutrophils as well, mediating resistance to nuclear factor-κB (NF-κB) inhibitors, as demonstrated in a KRAS-driven mouse tumor model [52].

Durrans et al. carried out a study to further elucidate the role of activated or re-programmed stromal cells within NSCLC specimens and their impact on tumor progression [68]. An increased number of bone marrow-derived cells of hematopoiesis were observed adjacent to and within the tumor stroma of NSCLC when compared to matched healthy lung tissue. Next, the authors compared the transcriptomes of myeloid compartments harbored by the NSCLC samples with non-malignant lung tissue stemming also from NSCLC patients. Notably, differentially regulated genes and mRNA sequences were found in the tumors, and certain molecules impacting gene function, such as osteopontin or chemokine (C-C motif) ligand 7 (CCL7), were present [68]. The conclusion drawn from this study is that homogeneous stromal fractions of myeloid cells are found in NSCLC, differing genetically from myeloid cells in the normal lung.

Osteocalcin-positive osteoblasts in the bone marrow of mice were systemically activated by the secretion of soluble receptors for an advanced glycosylation end product (sRAGE) by KRAS-driven lung adenocarcinomas [69]. The production of sialic acid-binding immunoglobulin-like lectin F (SiglecF) neutrophils was observed upon bone marrow activation, and this specific neutrophil subtype homed to the lungs, promoting tumor progression. Enhanced angiogenesis, myeloid cell differentiation, and the blocking of T cell functions were observed. The SiglecF neutrophil signature was also associated with an adverse outcome in patients suffering from lung cancer, thus indicating relevance not only in mice but also in humans [69].

Not only primary tumor growth but also the process of metastasis is linked to TAN function. Disseminated tumor cells from lung cancer have been found to extravasate into healthy tissue adjacent to the tumor, crossing endothelial barriers [70,71]. Consecutively, the disseminated tumor cells are colonized by a variety of mechanisms facilitating metastatic spread, including hypoxia [72], chemokine-dependent [73], and arachidonate 5-lipoxygenase (ALOX5)-dependent leukotriene synthesis [74] to recruit neutrophils. Furthermore, neutrophils endorse the proliferation and colonization of leukotriene B4 receptor 2 (LTB4R2)-positive disseminated tumor cells [74], thereby impacting metastasis. In pre-metastatic lung cancer, polarized neutrophils can suppress the activation of CD8+ T cells, blocking anti-metastatic immunity and thereby acting pro-carcinogenically [75]. Metastasis in the lung can be promoted by neutrophils secreting cathepsin G and neutrophil elastase, thereby degrading the anti-tumorigenic molecule thrombospondin 1 (TSP1) [76]. Moreover, metastasis is influenced by TANs via the atypical chemokine receptor 2 (ACKR2), expressed on hematopoietic progenitor cells. ACKR2 retains and regulates myeloid cell differentiation and function, and the direct targeting of host ACKR2 enhances the ability of neutrophils to specifically attack lung metastases, as shown in a breast cancer mouse model [77].

TANs also impact angiogenesis by a variety of mechanisms, thereby endorsing tumor growth, progression, and metastasis [78,79,80]. TANs up-regulate pro-angiogenic factors such as VEGF, most probably through STAT3 and c-Myc activation [81]. The up-regulation of proteins specifically promoting the course of metastasis has also been attributed to TANs, including, for example, Bv8, MMP9, S100A8, and S100A9 [81].

It must be pointed out that, in contrast to murine models where the phenotype and function of TANs have been studied, human cancers feature a slow and gradual evolution. By contrast, rapidly growing and highly immunosuppressive cancers have been observed in mouse models. Thus, it is crucial to investigate all stages of TAN interaction within the tumor microenvironment in humans especially, because different therapeutic anchor points may arise when looking at different stages of tumor development [82]. Eruslanov et al., therefore, compared the function of TANs in early-stage versus late-stage human lung cancer, shedding more light on TAN plasticity across all steps of tumor development. The results from this study comprise a distinct characterization of TANs in early-stage human NSCLC and the complex crosstalk between TANs and T cells. The authors found TANs to feature composite characteristics of neutrophils and antigen-presenting cells. More research needs to be done on the role of canonical TANs versus the antigen-presenting cell-like hybrid TANs, both in early-stage as well as in advanced-stage lung cancer. It may be this certain plasticity of TANs that renders them tumorigenic and especially aggressive, while they might—at early tumor stages—also exert an anti-tumorigenic function [82].

This study by Eruslanov et al. from 2017 aimed at characterizing TANs in early-stage human lung cancer [82]. One limit of the characterization of neutrophils within cancer tissue is the detection of only single or double granulocytic markers by immunohistochemistry. To overcome this obstacle, in this study, fresh tissue from surgical procedures of early-stage lung cancers was collected, and an extensive phenotypical analysis of TANs within the cancer tissue was performed. Multicolor flow cytometry was used to define TANs as CD11b+, CD15hi, CD66b+, MPOhi Arg1+, CD16int, and IL-5Ra cells. These TAN subtypes occurred within tumors with varying frequency, ranging from 2 to 20% of cells in the tumor microenvironment [2,82]. The neutrophils that had been recruited into lung tumors showed an activated phenotype, as compared to circulating peripheral blood neutrophils. This was shown by the expression of activation molecules, for instance, the up-regulated adhesion molecule CD54 (ICAM-1) and the downregulation of CD62L (L-selectin) and CD16 [67]. CD54, which is a mandatory molecule for cellular adhesion, endothelial transmigration, and the stabilization of interactions between cells, is also linked to TAN function. CD54, as recently demonstrated, is involved in IL-12 and IFN-γ-release by dendritic and natural killer cells [83]. CD54 was found to be a neutrophil marker of the N1 phenotype as well, as shown in mouse cancer models [7]. Eruslanov et al. also found activated TANs in lung cancer to express a new chemokine receptor profile in the tumor microenvironment [82]. When compared to neutrophils from the peripheral blood of lung cancer patients, TANs up-regulated CCR5, CCR7, CXCR3, CXCR4, and down-regulated CXCR1 and CXCR2. These chemokine receptors are usually upregulated in neutrophils at the sites of inflammatory processes, but also enhance the recruitment of neutrophils into tissues. Moreover, other neutrophil functions such as the release of α-defensin and consecutive bacterial cytotoxicity, as well as the respiratory burst in pulmonary neutrophils, are regulated by these specific chemokine receptors [84].

Co-stimulatory molecules, such as, for instance, CD86, CD54, OX40L, and 4-1BBL, are also expressed by TANs at a low level. Upon interaction with activated T lymphocytes, these co-stimulatory molecules are strikingly upregulated [2,67]. Quiescent neutrophils do not usually express such co-stimulatory molecules on their cell surfaces, but rather store them in cytoplasmic granules [85]. When activated, neutrophils transfer the stored molecules to their cell surface and are able to re-synthesize them [78]. In a subset of early-stage lung cancer patients, activated TANs expressing atypical cell surface markers, normally found only in antigen-presenting cells, were outlined [67,82].

As mentioned earlier, Carus et al. investigated the impact of both neutrophils and macrophages in 335 patients with resectable stage I–IIIA NSCLC [22]. This study showed that although tumor-associated CD66b(+) neutrophils and CD163(+) macrophages were correlated with adverse prognostic factors in NSCLC—such as a higher CRP, a higher white blood cell count, larger tumor size, and necrosis—no direct link to the patients’ outcome was observed [29].

## 4. Outlook: Anti-Cancer Vaccines as an Emerging Immunotherapeutic Tool and the Implications of TANs

The results on TANs in lung cancer summarized above highlight the vast effects of the host’s immune system and the tumor microenvironment in exerting a pro-carcinogenic, or else an anti-carcinogenic, effect. Tumor cells are only weakly immunogenic in general, which is one of the major reasons why cancers so easily undergo immune escape [86]. One example of how to make use of TANs in cancer therapy is the neoantigen vaccine approach. Increasing evidence suggests that vaccines alter neutrophil function long-term, as was shown, for example, in BCG vaccination [87]. Similarly, during immunization with the pneumococcal conjugate vaccine, adequate neutrophil function and activation are a prerequisite [88]. A mouse study showed, for instance, that neutrophils play a crucial role in the response to an experimental live attenuated Leishmania vaccine [89]. Immunized mice showed higher neutrophil recruitment to the vaccination site and adjacent lymph nodes when compared to controls [89]. Another study on bladder cancer, which is frequently treated with the BCG vaccine, showed that neutrophils are essentially linked to the anticarcinogenic effect of the BCG vaccine [90]. According to this study, there is a direct cytotoxic effect of neutrophils as well as a higher cytokine-, tumor necrosis factor (TNF) α-, interleukin (IL) 1β-, IL 6-, and TNF-related apoptosis-inducing ligand concentration at target sites upon BCG treatment. It is assumed that neutrophils endorse the recruitment of these proinflammatory molecules, thereby enhancing the anticarcinogenic effect of the BCG vaccine [90]. 

In recent years, research has increasingly focused on the investigation of neoantigens as therapeutic targets with greater immunogenicity and the aim to create novel immunotherapeutics. TANs, also being antigen-presenting cells in the tumor microenvironment, may also play a key role in the development of immunotherapeutic agents, i.e., the generation of neoantigen vaccines against cancers. Sequencing approaches, with the aid of bioinformatics, have been applied to investigate the action mechanisms of several MHC proteins in antigen presentation [91,92], with the subsequent cloning and expression of genes, leading to the in vitro culturing of antigen-specific T cells [93,94]. Thereby, neoantigens that may serve as the basis for anti-cancer vaccines can be identified. Tumor cells harbor certain neoantigens that render them immunologically different, as compared to healthy cells. In theory, neoantigen vaccines as a future type of immunotherapy may be used to induce a specific anti-tumor response in the host, generating a stable therapeutic effect [86].

Anti-tumor vaccines differ from canonical ones, since they are not prophylactically administered, but are given to patients with active malignant diseases with the intention of killing the tumor cells [95]. The mutational burden of tumor cells results in a change in the amino acid sequence of their proteins, which are translated and turned into short peptides. These peptides act as tumor neoantigens [96]. Conversely to autoantigens, tumor neoantigens have contact with MHC molecules, inducing an anti-tumor immune response in the host. The therapeutic effectiveness of tumor vaccines is linked to the aberrant expression of the targeted neoantigens between normal cells and cancer cells. T cells activated by the neoantigens of cancer cells can produce highly active T cells whose receptors feature a strong affinity towards MHC–neoantigen-peptide complexes and are less likely to be cleared by central immune tolerance [97].

Tumor driver mutations have a strong clonal tendency which is present in a large proportion of tumor cells. In a study by Schumacher et al., the mutation of isocitrate dehydrogenase type 1 (IDH1) was used to create an anti-tumor vaccine [98]. IDH1 mutation is observed in several cancer types, leading to aberrant enzyme function and malignant transformation. Mice were vaccinated with the IDH1 peptide, triggering an MHC-II-like effect and thereby an anti-tumor response [98]. Until today, different types of cancer vaccines have been created, comprising tumor cell vaccines [99], long peptide vaccines or protein vaccines [100,101], genomic vaccines [102], and dendritic cell-based vaccines [103,104].

The clinical application of anti-cancer vaccines is still at an early stage. Tumor-associated antigens are the main target of the canonical tumor vaccines that have been developed so far. These tumor-associated antigens are found in healthy as well as tumor cells [105]. Clinical trials of traditional anti-tumor vaccines targeting tumor-associated antigens have shown only limited success [105]. This can be explained by a gradual elimination of T cells that recognize tumor-associated antigens by the thymus, which renders the vaccine less effective over time. The establishment of neoantigen vaccines, which are based on tumor-specific mutations, showed more promising results in clinical studies. CD8+ and CD4+ cells within solid tumors can recognize neoantigens and can lead to an anti-tumor immune response in vivo [106,107]. In a study on patients suffering from malignant melanoma, Carreno et al. showed dendritic cells harboring neoantigens to effectively induce a T cell response and thus an anticarcinogenic effect in vivo as well [108]. Of the three melanoma patients enrolled in this trial, two had stable diseases and one showed no side effects or recurrence after vaccine therapy [108].

Tumor vaccines also exist based on RNA. RNA can be extracted from cancers and amplified for the generation of vaccines. Conversely to DNA-based vaccines, RNA vaccines are not integrated into the host’s genome. Sahin et al. established an RNA tumor vaccine using next-generation sequencing databases, creating a vaccine that would encode neoantigens that had previously been shown to be captured by dendritic cells [109]. Thirteen patients suffering from melanoma were treated with this RNA vaccine, and eight of them showed no tumor development at follow-up visits. In another clinical trial [110], a bioinformatic algorithm was used to predict the optimum combination of MHC-I molecules with neoantigens. Based on the bioinformatic results, a synthetic long peptide vaccine was created and six melanoma patients were administered the vaccine. No tumor recurrence for 32 months after vaccination was observed in four out of six patients [110]. Clinical studies on antitumor vaccines have also shown that, in patients with early-stage tumors, anti-cancer vaccines prove more effective as compared to patients with advanced diseases [111]. Summing up the clinical data that exists on anti-tumor vaccines so far, these vaccines have proven safe and moderately effective, inducing a CD8+- and CD4+-specific T cell response, but more research on this is warranted in the future.

In recent years, research has increasingly focused on the investigation of neoantigens as therapeutic targets with greater immunogenicity and with the aim of creating novel immunotherapeutics. TANs, also being antigen-presenting cells in the tumor microenvironment, may also play a key role in the development of immunotherapeutic agents, i.e., the generation of neoantigen vaccines against cancers. Sequencing approaches, with the aid of bioinformatics, have been applied to investigate the action mechanisms of several MHC proteins in antigen presentation [80,81], with the subsequent cloning and expression of genes, leading to the in vitro culturing of antigen-specific T cells [82,83]. Thereby, neoantigens that may serve as the basis for anti-cancer vaccines can be identified. Tumor cells harbor certain neoantigens that render them immunologically different, as compared to healthy cells. In theory, neoantigen vaccines as a future type of immunotherapy may be used to induce a specific anti-tumor response in the host, generating a stable therapeutic effect [79].

Anti-tumor vaccines differ from canonical ones, since they are not prophylactically administered, but are given to patients with active malignant diseases with the intention of killing the tumor cells [84]. The mutational burden of tumor cells results in a change in the amino acid sequence of their proteins, which are translated and turned into short peptides. These peptides act as tumor neoantigens [85]. Conversely to autoantigens, tumor neoantigens have contact with MHC molecules, inducing an anti-tumor immune response in the host. The therapeutic effectiveness of tumor vaccines is linked to the aberrant expression of the targeted neoantigens between normal cells and cancer cells. T-cells activated by the neoantigens of cancer cells can produce highly active T cells whose receptors feature a strong affinity towards MHC–neoantigen-peptide complexes and are less likely to be cleared by central immune tolerance [86].

Tumor driver mutations have a strong clonal tendency which is present in a large proportion of tumor cells. In a study by Schumacher et al., the mutation of isocitrate dehydrogenase type 1 (IDH1) was used to create an anti-tumor vaccine [87]. IDH1 mutation is observed in several cancer types, leading to aberrant enzyme function and malignant transformation. Mice were vaccinated with the IDH1 peptide, triggering an MHC-II-like effect and thereby an anti-tumor response [87]. Until today, different types of cancer vaccines have been created, comprising tumor cell vaccines [88], long peptide vaccines or protein vaccines [89,90], genomic vaccines [91] and dendritic cell-based vaccines [92,93].

The clinical application of anti-cancer vaccines is still at an early stage. Tumor-associated antigens are the main target of the canonical tumor vaccines that have been developed so far. These tumor-associated antigens are found in healthy as well as tumor cells [94]. Clinical trials of traditional anti-tumor vaccines targeting tumor-associated antigens have shown only limited success [94]. This can be explained by a gradual elimination of T cells that recognize tumor-associated antigens by the thymus, which renders the vaccine less effective over time. The establishment of neoantigen vaccines which are based on tumor-specific mutations, showed more promising results in clinical studies. CD8+ and CD4+ cells within solid tumors can recognize neoantigens and can lead to an anti-tumor immune response in vivo [95,96]. In a study on patients suffering from malignant melanoma, Carreno et al. showed dendritic cells harboring neoantigens to effectively induce a T-cell response and thus an anticarcinogenic effect in vivo as well [97]. Of the three melanoma patients enrolled in this trial, two had stable diseases and one showed no side effects or recurrence after vaccine therapy [97].

Tumor vaccines also exist based on RNA. RNA can be extracted from cancers and amplified for the generation of vaccines. Conversely to DNA-based vaccines, RNA vaccines are not integrated into the host’s genome. Sahin et al. established an RNA tumor vaccine using next-generation sequencing databases, creating a vaccine that would encode neoantigens that had previously been shown to be captured by dendritic cells [98]. Thirteen patients suffering from melanoma were treated with this RNA vaccine, and eight of them showed no tumor development at follow-up visits. In another clinical trial [99], a bioinformatic algorithm was used to predict the optimum combination of MHC-I molecules with neoantigens. Based on the bioinformatic results, a synthetic long peptide vaccine was created and six melanoma patients were administered the vaccine. No tumor recurrence for 32 months after vaccination was observed in four out of six patients [99]. Clinical studies on antitumor vaccines have also shown that in patients with early-stage tumors, anti-cancer vaccines prove more effective as compared to patients with advanced diseases [100]. Summing up the clinical data that exists on anti-tumor vaccines so far, these vaccines have proven safe and moderately effective, inducing a CD8+- and CD4+-specific T-cell response, but more research on this is warranted in the future.

To the best of our knowledge, TANs, as antigen-presenting cells within the tumor microenvironment, have so far not been used to generate cancer vaccines. We propose that, in the future, neoantigens presented by TANs may also be used to generate vaccines against malignant diseases. A combination of both—a neoantigen vaccine and specific recruitment of the antigen-presenting TAN phenotype in cancers—may be a promising therapeutic option as a novel approach to anti-tumor immune therapy.

## 5. Conclusions

TANs play a many-faceted role in lung cancer, as shown in murine models as well as in humans. The recruitment of neutrophils into the tumor stroma may have a pro-carcinogenic as well as an anti-carcinogenic effect.

Many details of the complex interplay of TANs in lung cancer with inflammatory molecules and tumorigenic pathways have been elucidated. Further research is warranted to clarify what key steps in tumorigenesis are influenced by TANs. The distinct characterization of TANs in human lung cancer is mandatory to outline further neutrophil signatures associated with prognosis and to outline future therapeutic targets or prognostic biomarkers.

Lastly, turning pro-carcinogenic TANs to anti-carcinogenic TANs may be an opportunity to boost natural anti-tumor immunity. In the future, even a vaccine-induced antitumor treatment may thereby be realized.

## Figures and Tables

**Figure 1 cancers-13-05972-f001:**
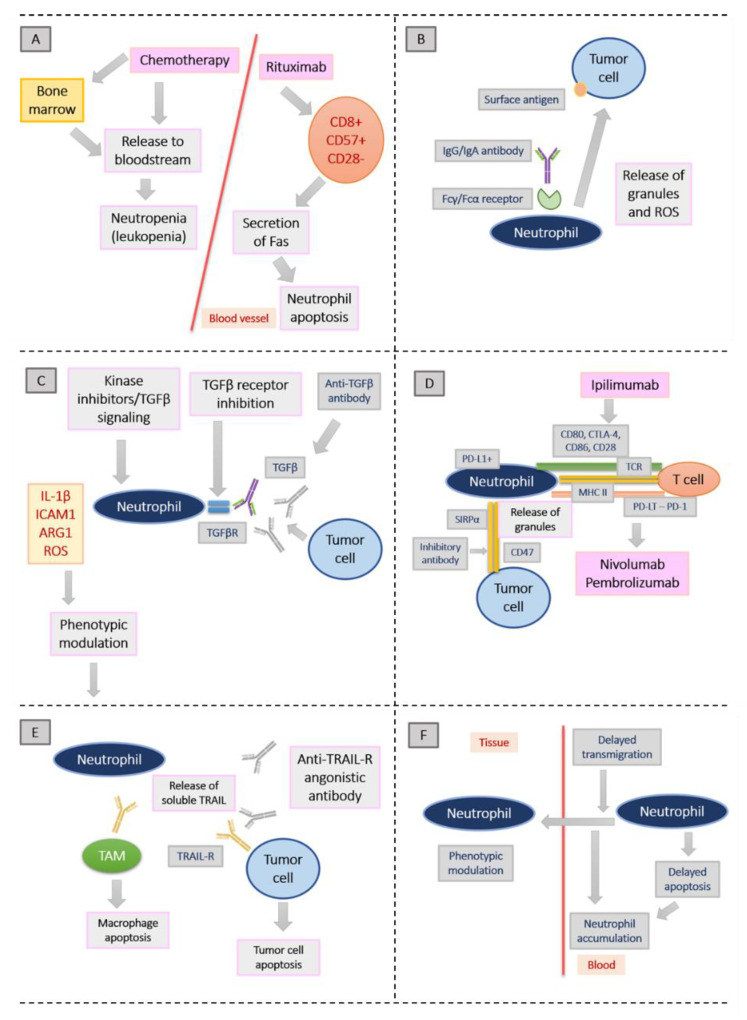
Possible mechanisms by which established anti-tumor treatments might modulate the phenotype and action mode of TANs. Figure adapted from Shaul et al. [56]. (**A**): Neutropenia is the result from the cytotoxic effect of chemotherapy, impairing bone marrow function and decreasing the amount of white blood cells released to the bloodstream. Fas secretion from T lymphocytes upon chemotherapy may also lead to neutrophil apoptosis. (**B**): After binding of monoclonal antibodies with an intact Fc domain to tumor cells, cellular toxicity is triggered. Neutrophils release their granules and lead to cancer cell apoptosis. (**C**): Transforming growth factor-β (TGFβ) signaling inhibition leads to the inhibition of its receptor (TGFβR), which in turn changes neutrophils towards an antitumor phenotype in mice. (**D**): Tumor immune escape is promoted by the activation of immune checkpoint inhibitors. Among others, cytotoxic T lymphocyte-associated protein 4 (CTLA-4), programmed cell death 1 (PD-1) and CD47 count among these checkpoint inhibitors. (**E**): Tumor necrosis factor-related apoptosis-inducing ligand (TRAIL) promotes cancer cell death by binding to TRAIL receptors (TRAIL-Rs) on their cell surfaces. Neutrophils are capable of releasing soluble TRAIL. (**F**): Neutrophils from the bloodstream accumulate as an effect of steroids, because their recruitment to tissues is inhibited. It is assumed that certain steroids may specifically modulate neutrophil phenotypes [56].

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
