# Peer review of "Role of Tumor-Associated Neutrophils in the Molecular Carcinogenesis of the Lung"

_cancers, 2021, doi:10.3390/cancers13235972_

Round 1

Reviewer 1 Report

In the current review Taucher et al give an overview on the  impact of TAN on lung cancer. They first describe the roles of TAN in cancer in humans and mice, and then focus on TANs in lung cancer, with emphasis on future potential related treatments.

The authors made a thorough review highlighting the divers and sometimes contradicting role of TAN in lung cancer. The review is basically divided to 3 – a general part on TANs, a specific part on TANs in Lung cancer, and an outlook on vaccines. The first two parts are broad, detailed, and relevant – much less so the third part (see below).

Major Comments:

  1. The authors nicely focus on molecular mechanisms of the effects of TANs. I think this should be emphasized further, maybe even in the title.
  2. The way the review is built is confusing and not accurate. There is a very long "Introduction" followed by "Review of the literature". Would suggest rearranging this, as it is indeed in the text. I suggest starting with a short introduction, followed by:
    * TANs in general
    * TANs in Lung cancer
    * Future perspectives and therapeutic implications
    * Conclusions
  3. Throughout the manuscript (Mainly in the TAN's section) the separation between murine and human data is not sufficient. It is important to better clarify what is human and what is murine data, and if possible – separate each "section" between the two. This is also needed in paragraph 2.1 that is going from mice to humans and back, and should be better separated.
  4. The paragraph on prognostic importance on TANS in different cancers (Page3 Line102-113) is probably better early in this section, before talking about localization.
  5. Murine models of TANs and potential translation to human cancers – nice and important paragraph. Possibly before starting talking about specific molecular pathways (P3 L130 and further) an introduction of the topic – molecular pathways in neutrophils, is warranted. Possibly part of it could be moving the paragraph in P4 L169-175 to an earlier place – before the detailed examples, and adjusted accordingly. 
  6. All the paragraph on vaccines – P9 bottom-P10 – very long and not very relevant to the topic of the review. This is a general description of cancer-vaccines – Should be probably shortened and better connected to the topic how are all these vaccines related to TAN?
    For example - the role of describing the study by Schumacher et al in details (Ref 87) is not clear. Is this a general example of anti-cancer vaccines? How is this vaccine related to TAN? If it is- more details are needed, If not- much less details… In addition anti-cancer vaccines are only one option of targeting TAN and should be described as such (an example), and possibly some other options of targeting TAN can be mentioned. Also - what are the combinations suggested in P10 L492-7 – not clear.

Minor Comments

  1. TAN form a subgroup of tumor-associated myeloid cells (Abstract) – not accurate, please rephrase.
  2. The sentence – "While, among myeloid tumor-infiltrating cells, tumor-associated macrophages have been investigated in-depth, the molecular impact of tumor-associated neutrophils (TANs) remains to be elucidated in detail" in the Introduction is somewhat exaggerated… It Is true that more research has been done on macrophages, but lots of research was done on neutrophils – see for example this review. Please rephrase.
  3. "…however, a proportion of cases is not related to smoking or any other common risk factor" - needs a reference and a relevant number (about 15%?)
  4. "…the accurate understanding of lung 48 cancer microbiology is crucial to develop new effective treatment agents" – Not clear – How is microbiology related here? are they talking on bacteria? Immune cells? [P2, L48-9]
  5. Not sure the word dichotomous (p3 L124) is right. The borders in TANs functions are actually not very clear. Suggest changing.
  6. "TANs are also found in the peripheral blood of patients suffering from breast cancer before undergoing surgery, (p5 L20-9) – not clear – TANs in the circulation? Not what the paper says.
  7. Figure 1 is adapted – was permitted by the original journal?
  8. The study by Carus et al (Ref 22) is described in details twice. Would suggest erasing the 2nd appearance at top P9.
  9. I think the paragraph on the study by Ilie et al on the prognostic effect of CD66b+ neutrophils (P8 L 398 to p9 L413) should come much earlier – at the beginning of the discussion on TANs in lung cancer.
  10. Several typos (?):

- Tough (p3 L129) – Typo? Not clear.

- G1+ (P4 L176-186) – shouldn't it be GR1?

- In this study X2 in P8 lines 368-9 (Typo)

- CARUS – mistake in P9L419

Author Response

Reviewer #1

In the current review Taucher et al give an overview on the  impact of TAN on lung cancer. They first describe the roles of TAN in cancer in humans and mice, and then focus on TANs in lung cancer, with emphasis on future potential related treatments.

The authors made a thorough review highlighting the divers and sometimes contradicting role of TAN in lung cancer. The review is basically divided to 3 – a general part on TANs, a specific part on TANs in Lung cancer, and an outlook on vaccines. The first two parts are broad, detailed, and relevant – much less so the third part (see below).

Major Comments:

The authors nicely focus on molecular mechanisms of the effects of TANs. I think this should be emphasized further, maybe even in the title.

Response: According to this input, we changed the title to “Role of tumor-associated neutrophils in molecular carcinogenesis of the lung.”

The way the review is built is confusing and not accurate. There is a very long "Introduction" followed by "Review of the literature". Would suggest rearranging this, as it is indeed in the text. I suggest starting with a short introduction, followed by:

* TANs in general

* TANs in Lung cancer

* Future perspectives and therapeutic implications

* Conclusions

Response: We thank Reviewer #1 for pointing out the confusing structure of this manuscript. Indeed, we think Reviewer #1 is right, and it makes sense to re-structure the article.

Thus, we have now arranged headings and sub-headings more comprehensively, namely:

*1. Introduction

*2. General aspects

*2.1 Tumor-associated neutrophils – what do we know?

*2.2 Murine models of TANs and potential translation to human cancers.

*3. TANs in lung cancer.

*4. Outlook: Anti-cancer vaccines as an emerging immunotherapeutic tool and implications of TANs.

*5. Conclusion

Throughout the manuscript (Mainly in the TAN's section) the separation between murine and human data is not sufficient. It is important to better clarify what is human and what is murine data, and if possible – separate each "section" between the two. This is also needed in paragraph 2.1 that is going from mice to humans and back, and should be better separated.

Response: We thank Reviewer #2 for the accurate and careful revision of our manuscript. We went through chapter 2.2 “Murine models of TANs and potential translation to human cancers.” again, but all articles reviewed in this chapter are about murine models. Some of them are about findings in humans as well, however, we aimed at discussing potential translation from murine to human carcinogenesis as well, so we think the structure of this chapter is appropriate.

The paragraph on prognostic importance on TANS in different cancers (Page3 Line102-113) is probably better early in this section, before talking about localization.

Response: Reviewer #2 is completely right, and accordingly, we have switched the paragraph on the prognostic importance of TANs to the beginning of this section (pages 2 and 3, lines 75 to 94), so that the paragraph about localization follows later.

Murine models of TANs and potential translation to human cancers – nice and important paragraph. Possibly before starting talking about specific molecular pathways (P3 L130 and further) an introduction of the topic – molecular pathways in neutrophils, is warranted. Possibly part of it could be moving the paragraph in P4 L169-175 to an earlier place – before the detailed examples, and adjusted accordingly.

Response: As suggested, we have moved the respective paragraph to an earlier place of this chapter (page 4, lines 139 to 145). Furthermore, we have added an introductory paragraph to this chapter (page 3, lines 121 to 129).

All the paragraph on vaccines – P9 bottom-P10 – very long and not very relevant to the topic of the review. This is a general description of cancer-vaccines – Should be probably shortened and better connected to the topic how are all these vaccines related to TAN?

For example - the role of describing the study by Schumacher et al in details (Ref 87) is not clear. Is this a general example of anti-cancer vaccines? How is this vaccine related to TAN? If it is- more details are needed, If not- much less details… In addition anti-cancer vaccines are only one option of targeting TAN and should be described as such (an example), and possibly some other options of targeting TAN can be mentioned. Also - what are the combinations suggested in P10 L492-7 – not clear.

Response: We understand that the paragraph on neoantigen vaccines might look a little “off topic”. We understand that the link to TANs is maybe not so clear at first glance.

We have added the sentence “One example how to make use of TANs in cancer therapy is the neoantigen vaccine approach.” – emphasizing that neoantigen vaccines are only one among many approaches of cancer immune therapy.

In order to give a better explanation, how this chapter is linked to TANs, we have included a new introductory paragraph (page 10, lines 436 to 450)

Minor Comments

TAN form a subgroup of tumor-associated myeloid cells (Abstract) – not accurate, please rephrase.

Response: We have rephrased this sentence: “Tumor-associated neutrophils (TANs), among tumor-associated macrophages and myeloid-derived suppressor cells, count among tumor-associated myeloid cells.” (see abstract)

The sentence – "While, among myeloid tumor-infiltrating cells, tumor-associated macrophages have been investigated in-depth, the molecular impact of tumor-associated neutrophils (TANs) remains to be elucidated in detail" in the Introduction is somewhat exaggerated… It Is true that more research has been done on macrophages, but lots of research was done on neutrophils – see for example this review. Please rephrase.

Response: We thank Reviewer #1 for pointing this out. Hence, we have rephrased this sentence: “In the past, more research has been done on tumor-associated macrophages than on tumor-associated neutrophils (TANs). However, in the last decade TANs have been increasingly investigated and the underlying molecular mechanisms of TANs are gradually elucidated.” (page 1, lines 36 to 39)

"…however, a proportion of cases is not related to smoking or any other common risk factor" - needs a reference and a relevant number (about 15%?)

Response: This is indeed a good remark. We have now given the percentage (10-15%) and added a citation (page 2, lines 51 to 54).

"…the accurate understanding of lung 48 cancer microbiology is crucial to develop new effective treatment agents" – Not clear – How is microbiology related here? are they talking on bacteria? Immune cells? [P2, L48-9]

Response: Reviewer #1 is right, this sentence is somewhat confusing. We have changed it to “(…) the accurate understanding of molecular lung carcinogenesis and immunological aspects in particular, (…)” (page 2, lines 52 to 54)

Not sure the word dichotomous (p3 L124) is right. The borders in TANs functions are actually not very clear. Suggest changing.

Response: Accordingly, we have changed the word to “pivotal” (as used in other articles in this context as well). (page 4, line 146)

"TANs are also found in the peripheral blood of patients suffering from breast cancer before undergoing surgery, (p5 L20-9) – not clear – TANs in the circulation? Not what the paper says.

Response: Indeed, this is what the paper says. In this article, they call TANs “Tumor-entrained neutrophils (TENs)”, which are different from canonical neutrophils of the peripheral blood when characterized molecularly. For reference, see Granot et al., Cancer Cell, 2012.

Figure 1 is adapted – was permitted by the original journal?

Response: We obtained permission from the author of this article. Therefore, we have added an acknowledgement. (line XX)

The study by Carus et al (Ref 22) is described in details twice. Would suggest erasing the 2nd appearance at top P9.

Response: We thank Reviewer #1 for pointing this out. The second appearance was therefore greatly shortened (page 10, lines 424 to 429).

I think the paragraph on the study by Ilie et al on the prognostic effect of CD66b+ neutrophils (P8 L 398 to p9 L413) should come much earlier – at the beginning of the discussion on TANs in lung cancer.

Response: Reviewer #1 is right, so we have shifted the paragraph about the paper by Ilie et al. upwards to chapter 3 “TANs in lung cancer.” (page 7, lines 294 to 309)

Several typos (?):

- Tough (p3 L129) – Typo? Not clear.

- G1+ (P4 L176-186) – shouldn't it be GR1?

- In this study X2 in P8 lines 368-9 (Typo)

- CARUS – mistake in P9L419

Response: We thank Reviewer #1 for the careful revision. All these typos have been corrected (changes are marked in red).

Reviewer 2 Report

This is an academic and very well researched review. The role of Tumor associated neutrophils in tumors in general and in lung cancer in particular have been explained in detail.The concept of anticarcinogenic and procarcinogenic  neutrophils has been delineated in fine detail.

The highlight of the work is expressed in furure therapeutic options for the functional conversion of procarcinogenic TANs to anti antitumorogenic.

Author Response

Reviewer #2

This is an academic and very well researched review. The role of Tumor associated neutrophils in tumors in general and in lung cancer in particular have been explained in detail.The concept of anticarcinogenic and procarcinogenic  neutrophils has been delineated in fine detail.

The highlight of the work is expressed in furure therapeutic options for the functional conversion of procarcinogenic TANs to anti antitumorogenic.

Response: We thank Reviewer #2 for the revision of our manuscript. We are grateful for the good review.

Round 2

Reviewer 1 Report

The authors have convincingly addressed almost all my comments. 
Please note only the following very minor comments:

  • In the abstract - "Tumor-associated neutrophils (TANs), among tumor-associated macrophages and myeloid-derived suppressor cells, count among tumor-associated myeloid cells." I think the first among was meant to be "together with"?
  • In the authors response - "Indeed, this is what the paper says. In this article, they call TANs “Tumor-entrained neutrophils (TENs)”, which are different from canonical neutrophils of the peripheral blood when characterized molecularly. For reference, see Granot et al., Cancer Cell, 2012." I am very familiar with this paper "(not mine...). TENs are circulating neutrophils that were entrained by factors from the tumor, and are not TAN. The term TAN refers to neutrophils inside the tumors.

Author Response

Response to Reviewer #1

The authors have convincingly addressed almost all my comments. 
Please note only the following very minor comments:

  • In the abstract - "Tumor-associated neutrophils (TANs), among tumor-associated macrophages and myeloid-derived suppressor cells, count among tumor-associated myeloid cells." I think the first among was meant to be "together with"?

Response: Of course, Reviewer #1 is right. The first “among” was meant to be “together with”. We have changed this mistake, and we thank Reviewer #1 again for the very careful revision of our work.

  • In the authors response - "Indeed, this is what the paper says. In this article, they call TANs “Tumor-entrained neutrophils (TENs)”, which are different from canonical neutrophils of the peripheral blood when characterized molecularly. For reference, see Granot et al., Cancer Cell, 2012." I am very familiar with this paper "(not mine...). TENs are circulating neutrophils that were entrained by factors from the tumor, and are not TAN. The term TAN refers to neutrophils inside the tumors.

Response: Also here, Reviewer #1 is right. We have unfortunately mixed up the terms “TEN” and “TAN”. We have deleted the whole sentence “Interestingly, TANs are also found in the peripheral blood of patients suffering from breast cancer before undergoing surgery, however, not in healthy subjects (17).”